# A Hardware-in-the-Loop V2X Simulation Framework: CarTest

**DOI:** 10.3390/s22135019

**Published:** 2022-07-03

**Authors:** Jian Wang, Yu Zhu

**Affiliations:** College of Computer Science and Technology, Jilin University, Changchun 130012, China; zhuyu20@mails.jlu.edu.cn

**Keywords:** V2X, simulation testing, HIL challenges and solutions, ICW, CACC

## Abstract

Vehicle to Everything (V2X) technology is fast evolving, and it will soon transform our driving experience. Vehicles employ On-Board Units (OBUs) to interact with various V2X devices, and these data are used for calculation and detection. Safety, efficiency, and information services are among its core uses, which are currently in the testing stage. Developers gather logs during the real field test to see if the application is fair. Field testing, on the other hand, has low efficiency, coverage, controllability, and stability, as well as the inability to recreate extreme hazardous scenarios. The shortcomings of actual road testing can be compensated for by indoor testing. An HIL-based laboratory simulation test framework for V2X-related testing is built in this study, together with the relevant test cases and a test evaluation system. The framework can test common applications such as Forward Collision Warning (FCW), Intersection Collision Warning (ICW) and others, as well as more advanced features such as Cooperative Adaptive Cruise Control (CACC) testing and Global Navigation Satellite System (GNSS) injection testing. The results of the tests reveal that the framework (CarTest) has reliable output, strong repeatability, the capacity to simulate severe danger scenarios, and is highly scalable, according to this study. Meanwhile, for the benefit of researchers, this publication highlights several relevant HIL challenges and solutions.

## 1. Introduction

Intelligent Transportation Systems (ITS) have been evolving at a rapid pace in recent years. Vehicle to Everything (V2X) technology is also developing, with the benefit of being able to perceive where cameras and radar cannot detect, compensating for automated cars’ perceptual blind spots. The essential components of ITS [1] are shown in Figure 1. Vehicles have OBU that broadcast their own messages, such as a Basic Safety Message (BSM), which carries information such as the vehicle’s driving condition. By receiving messages from nearby OBUs and Road Side Units (RSUs), the OBU senses the surroundings and uses this information to run applications (e.g., various types of alerts, CACC and so on) [2].

Vehicles in ITS are equipped with an OBU, which consists of a positioning system, a radio communication subsystem, and an on-board unit that wraps its status information in a BSM message and broadcasts it. The positioning system and the in-vehicle bus are the primary data sources, with the positioning system providing vehicle location and motion status information (e.g., latitude, longitude, speed, acceleration, etc.) and the in-vehicle bus (mostly the Controller Area Network (CAN) bus) providing other status information (e.g., speed, acceleration, brake status, turn signal status, etc.). Simultaneously, the OBU receives V2X messages via the radio communication subsystem and delivers the specific application over CAN or Local Area Network (LAN) to the Human Machine Interface (HMI) [3], as shown in Figure 2. The OBU receives GPS as well as V2X information through the antenna. The antenna interface can also be connected directly to the signal generator.

OBU’s application is currently in its early stages of development. Drivers will be misled by incorrect warning notifications, which will have an impact on their normal driving. As a result, the generated function module must be tested to ensure that it functions properly and is in good working order.

In a field test, the Host Vehicle (HV) and the Remote Vehicle (RV) can be organized using an OBU to execute simulations to test their functionality. It is, however, less efficient and reproducible, and it cannot represent risky events (e.g., impending collision, having collided). Furthermore, collecting test logs for real-world vehicle testing is challenging, and automated analysis and assessment of test logs is tough to perform. While real-world testing is crucial, simulation testing may help enhance testing speed and quality. In the case of Intersection Collision Warning (ICW), as an HV approaches a junction, there may be obscured visibility and a limited sensor sensing range. To increase junction access safety, V2X could collect data from side-tracking cars, compute whether vehicles are at danger of collision, and notify the driver [4]. On account of the inaccurate warning timing and trigger conditions of ICW algorithms, drivers may receive incorrect warnings, impacting their driving and possibly causing traffic accidents. Due to the high cost of field testing, a laboratory testing framework is needed to conduct a V2X communication simulation, automated testing, and test result assessment.

A Hardware-in-the-Loop (HIL) V2X simulation framework (CarTest) is proposed in this study. To increase testing efficiency and assist the development of V2X applications, it creates authentic test scenarios in the simulation engine, generates appropriate test cases, translates the data in the simulation to the necessary hardware devices, and records test results as well as test logs for review.

## 2. Related Work

To achieve a high degree of safety, the development of V2X technology necessitates regular verification and testing of functioning under diverse driving circumstances. Many simulation test tools for autonomous driving, such as Veins, iTETRIS, and VSimRTI [5,6,7], have increasingly added V2X functional testing. They primarily advocated for the potential of testing V2X applications using simulation test software and made suggestions for simulation test software selection. However, their work involves software-in-the-loop testing, which is separate from the hardware, and it is tough to disagree about their V2X apps’ real performance.

mboxOn the other hand, HIL testing can further improve the accuracy of testing. Wang, J. et al. [8] studied and summarized the virtual-real testing method in terms of the needs and challenges of V2X applications and testing requirements. Gelbal, Y. et al. [9,10,11,12] constructed an HIL testing system and evaluated the lane maintaining, Adaptive Cruise Control (ACC) algorithms, and pedestrian collision warning algorithms. Its assessment capabilities, whereas, is limited, and it is unable to run a huge number of tests. In addition, numerous hardware constraints in HIL testing have yet to be resolved. mboxFurther, Chen, S. et al. [13,14,15,16] used OBU and Electronic Control Unit (ECU) as part of a simulation platform to improve the efficiency of development and testing. Many algorithms such as trajectory planning and control were verified using these systems. Zhang, E. et al. [16,17,18,19] enhanced the evaluation capability of the testbed. However, their OBU tests are all small in number and only guarantee functional tests in environments with good communication quality, but not in congested environments.

In conclusion, all associated work on V2X test simulation testing has been performed; however, there is no one universal answer. The software and hardware in the aforementioned test framework are too tightly connected to allow for software and hardware change. The test function is rather simple, and the test framework can only test one function; therefore, all OBU functions cannot be tested using a single test platform. CarTest is a typical testing and evaluation platform for V2X applications.

## 3. System Model

This chapter will introduce the basic components of CarTest, as well as the existing challenges and possible solutions. The CarTest mentioned in this article is our independently developed software that will eventually be made available under the GNU General Public License.

### 3.1. Framework

We discussed the OBU’s communication mechanism as well as its electrical and electronic surroundings in the preceding section. CarTest, an HIL-V2X simulation framework, was created after evaluating OBUs from multiple manufacturers. The gray section in the illustration is the replaceable part, which is compatible with various software and hardware due to the interface design, as shown in Figure 3.

The traffic scenario simulation engine is used to simulate traffic scenarios. We created a collection of test cases for various V2X applications. CarTest also offers a collection of automated testing tools that can run tests automatically when test cases are selected. The data packing module maps the simulation engine’s host vehicle to the Device Under Test (DUT), as well as the rest of the scenario’s objects (such as distant automobiles, road signs, and so on) to a standard OBU or signal generator. If OBU is employed, it must pass specific tests to ensure that its transceiver performance is flawless. CarTest use GNSS emulator, CAN emulator and channel emulator to achieve the overall HIL of OBU. During testing, logging and application outputs (such as FCW, ICW, and so on) are recorded, and the application outputs are presented on the HMI. Individual test cases are assessed simultaneously in real time, yielding test results. An overall test report is generated when all test cases have been finished. Test logs, assessment findings, and data visualization capabilities are all included in the test report. V2X application developers can use the test findings to improve their individual apps. In this paper, the OBU-equipped vehicle is defined as HV, the nearby driving vehicle is RV, and the device under test is DUT. its data interaction diagram is shown in Figure 4.

The simulation engine starts once the test begins and explores the test set automatically. Logging and data transmission will both take place at the same time. After the first test case is completed and assessed using the current test logs, the second test case is performed. An overall test report is created and may be seen by testers once all test cases are finished, as shown in Figure 5.

### 3.2. GNSS Simulation

For the simulation of GNSS information from the host vehicle, the framework employs a GNSS signal generator. The data packaging module encapsulates data from the simulation engine, and the signal generator generates an RF signal that is linked to the DUT’s GNSS interface, as shown in Figure 6.

Universal Transverse Mercator Grid System (UTM) coordinates are planar right-angle coordinates, and this coordinate grid system and the projections based on which have been widely used in topographic maps, as a reference grid for satellite imagery and natural resource databases, and in other applications where precise positioning is required. In the UTM system, the surface area of the Earth between 84° N and 84° S is divided into north–south longitudinal bands (projection bands) by 6° of longitude. These projection bands are numbered from 1 to 60 starting at 180° longitude and moving eastward. Each band is further divided into quadrilaterals with a latitudinal difference of eight degrees. When the numbers are too large, it is also possible to add a fixed offset to the UTM coordinates to make data processing easier.

It’s worth noting that the XYZ coordinate system is used as the reference coordinate system in the simulation scenario files. There are two different sorts of simulation scenarios. The first is a reproduction of a genuine landscape, and it is advised that the World Geodetic System 1984 (WGS84) coordinate system be converted to UTM coordinate system directly using the Proj package. The second method involves creating a virtual scene, such as a fictional junction, mapping a point to the appropriate XYZ coordinate system, and using Geodesic themes [20] to solve the coordinates of all points. The particular procedure is depicted in Figure 7.

The origin of the coordinates in the simulation map is taken as far as possible to the lower left of the test area. This ensures that the test area is in the first quadrant of the XY coordinate system, which can alleviate some of the work, as shown in Figure 7.

For example, take the starting point in the simulation engine as (20, 5, 0) and map it to (3,380,679, 789,883, 150) in the UTM coordinate system. Based on the above conversion we can set a UTM coordinate offset = (3,380,000, 790,000, 150). Then the point can be expressed as (679, −117, 0), which can make the numerical representation more intuitive, as shown in Equation (Equation 1).
(1)$$\left(20,\;5,\;0\right)\overset{map}{⟶}\left(338,0679,\;789,883,\;150\right)\overset{offset}{⟶}\left(679,\;-117,\;0\right).\;$$

The units of coordinate system for XYZ and UTM are meters, so their conversion method is relatively simple. If the vehicle moves 1000 m along the X-axis, the vehicle is currently located at point (1020, 5, 0). This point can be mapped to point (3,381,679, 789,883, 150) of the UTM coordinate system. After deflection, point (1679, −117, 0) is obtained, as shown in Equation (Equation 2).
(2)$$\left(20,\;5,\;0\right)\overset{move}{⟶}\left(1020,\;5,\;0\right)\overset{map}{⟶}\left(3,381,679,\;789,883,\;150\right)\overset{offset}{⟶}\left(1679,\;-117,\;0\right).\;$$

In this paper, Using VTD as a simulator, GNSS simulation results are shown in Figure 8. On the right, you can see the map’s top view. The simulator’s model is exhibited at the bottom of the left side, with the historical track in orange and the recent motion track in blue.

The analog signal can be received after the test. However, it is worth noting that:When the test case is altered, the geographic location of the test case will change, and there will be no ephemeris file for the current map, resulting in signal loss. As a result, the test platform, which is detailed in the experimental section, is utilized to test in this case;Clock synchronization and delay situation: The ring’s hardware demands a high level of clock synchronization and must keep the LAN environment running smoothly.

### 3.3. Can Simulation

On CAN, a large amount of data are exchanged, whereas DUT just requires a small amount of important data. As indicated in Table 1, HV’s CAN data injection test primarily covers vehicle motion information (e.g., speed, acceleration, etc.) and vehicle status information (e.g., turn signal status, brake status, etc.). Because the needed Database Can file (dbc file) varies depending on the type of OBU, this framework proposes a dbc file to fulfill the fundamental demands of the test and assure a certain degree of adaptability.

The data encapsulation module encrypts the data and injects them into the DUT through the CAN signal generator. When injecting CAN signals in the OBU, it is important to note that:Is there a wake-up frame on the DUT?;The CAN signal’s operational frequency.

### 3.4. V2X Simulation

The data packaging module obtains the information of all the distant vehicles in the simulation engine and packages the messages of each vehicle into BSM. finally, the signal simulation is performed by OBU or Signal Generator, and the channel fading simulator can also be added to simulate the real channel environment (e.g., countryside environment, high-speed environment, etc.). Two V2X model simulation schemes are proposed in this paper, as shown in Figure 9.

V2X signal generator: CMW500 is currently being used as an RF signal generating source [21], and it is capable of transmitting RF signals. A signal generator can generate BSM for up to 100 vehicles in real time. However, the signal goes to the same RF port, so the simulated signal is the similar one. C-V2X is still in its early stages of development, with several revisions to its physical layer, access layer, and other components. The solution is not adaptable to the development environment and is not versatile;OBU as signal generator: It is also feasible to employ a specific OBU that’s been demonstrated to work well as a signal generator. Theoretically, the more OBUs deployed, the more simulated vehicles can be performed. However, the communication frequency of V2X is 10 HZ, and when the number of vehicles is greater than 200, the data bus of the simulator will be under greater pressure. So at the same moment, a maximum of 200 vehicles are supported in a normal test environment. However, we do not recommend deploying too many OBUs for HIL testing because the fading simulator cannot handle too many signals. When it is necessary to test more complex channel environment, we recommend using large-scale testing, which can simulate stronger interference signals by increasing the number of OBU.This technique is more cost effective when simulating a small number of automobiles. The solution is Abstract Syntax Notation One (ASN.1) switchover adaptable, extendable, and secondary development friendly.

The fading simulation is an optional component. Different road conditions correspond to various channel environments, with the urban environment being the most complicated. The channel simulator makes the signal as near to the real-world electromagnetic signal as feasible [22].

### 3.5. Test Cases Library

There are 17 main V2X applications [23] that build a test case library, as shown in Table 2. The process of building test cases: firstly, a test case template is built manually, and then sub-test cases are derived automatically by generating different speed conditions of vehicles to ensure the coverage of test cases.

As an example, the ICW is utilized, and the traditional three test cases are listed in Figure 10. The blue vehicle is the HV, and the red vehicle is the RV, and the two vehicles are traveling at a constant speed, with the collision risk varying depending on the speed combination. The test cases that are at risk of colliding must be informed, whereas others do not. It is feasible to tell whether the algorithm passes based on the DUT’s warning state. A vast number of test cases can be used to evaluate the algorithm’s strengths and drawbacks.

### 3.6. Indoor Testing Setup

The indoor test setup mainly includes GNSS as well as LAN environment, as shown in Figure 11.

First, indoor test needs to ensure successful OBU positioning. Whether using actual GNSS satellite signals, or satellite signals generated by a signal generator, An indoor GNSS amplifier needs to be set up, which can ensure that all OBUs in the lab can get the positioning information quickly. The amplifier is also recommended to be placed on the ceiling of the laboratory. If an actual satellite signal is used, a receiver needs to be installed on the roof of the building to connect to the indoor amplifier. Second, indoor testing is recommended to use industrial routers to organize Wireless Local Area Network (WLAN). The latency is guaranteed to be less than 10 ms at 200 devices. Finally, CarTest is deployed on the server. After starting the test, OBU will connect to the testbed. At the beginning of the test, control commands will be sent down to the OBU via WLAN. Part of the status information will be reported by OBU and recorded in OBU at any time during the test. At the end of the test, the OBU will automatically upload the log records to the server, as shown in Figure 12.

During testing, some test programs need to be installed on the OBU side for receiving control commands, data upload, and log upload. Control commands include the setting of parameters for OBU communication, and whether the application is on is encoded in protobuf format. Message Queuing Telemetry Transport (MQTT) performs better in multi-device, high frequency Internet of Things (IoT) communication environment. Real-time reporting of logs uses MQTT-EMQX as middleware. Redis will be used as a data cache queue that will be progressively persisted into MySQL. In indoor tests, WLAN is better, as it guarantees a delay of less than 10 ms. In the outdoor test, the coverage performance of 4G is better due to the larger test area, but the delay should not exceed 30 ms.

### 3.7. Field Testing Setup

Outdoor testing requires consideration of the placement of the OBU and the power supply method. Severe weather conditions (e.g., high temperature, rainfall, etc.) must also be taken into account. Therefore, we designed an outdoor test trolley, which can place 8 OBU and including water stopper, equipped with battery pack. The height of the antenna tray on this trolley is 1.5 m because the antenna of a typical vehicle is mounted on the roof of the vehicle (approximately 1.5 m). The OBU is placed on top of the heat sink baffle, and the battery as well as cables are placed below the baffle, as shown in Figure 13.

At the same time, the outdoor test communication range is large, and it is recommended to use 4G network to complete the control of OBU.For testing, the OBU is mounted on a trolley and placed in the road. Multiple trolleys can be placed to simulate complex or strong signal interference channel environments.

## 4. Case Study

This section will introduce the test page and hardware deployment of CarTest. Using CarTest, we have conducted ICW test, Collaborative Adaptive Cruise Control (CACC) test, large-scale test, GNSS test to verify its testing capability.

### 4.1. Test Platform

The test portal provided by CarTest is a test platform online, and the test flow is represented in Figure 5. As shown in Figure 14, the test case administration interface contains add, delete, and check features as well as test control. Testers can choose from a list of pending test cases or import existing test scenarios. Testers can name the task, specify the test type and ASN.1 version, and then choose “Start testing” or “Save as plan”.

Compared with other testing platforms, CarTest has the following advantages:B/S structure, which can realize cloud simulation and be operated by testers using laptops;Interface-based design, so it can be compatible with CARLA, VTD, panosim, etc;Rich test cases and perfect management functions;Support long time and large scale testing;Perfect evaluation system and visualization of evaluation results;The CAN signal’s operational frequency.

An automated test script then takes over, runs each test case one by one while logging different data (such as HV, RV information, warning messages, and so on) and assessing the test case based on the logs. The various tests are detailed below.

### 4.2. ICW Test

A fundamental feature is intersection collision warning. An HV driving straight through an intersection and an RV entering into that lane from a side lane are the test cases. The test platform listens for the warning signal as the RV approaches the junction and shows it in the HMI. The test passed because it produced the expected outcome, as illustrated in Figure 15.

When creating a test case, we make a note of the expected warning value. 0 indicates that no warning should be sent, whereas “0x 0101” indicates ICW. It is evaluated to pass if the warning result has the intended warning value. The outcome is represented in the following equation, which uses W to symbolize the set of received warning values (e.g., 257,258) and $W_{E}$ to denote the set of expected warning values (e.g., 258).
(3)$$W\cap W_{E}\neq ⌀?\;\mathrm{pass}:\;\mathrm{fail}\;.$$

The ICW function has been subjected to several testing. The pass rate for each of the 1160 use cases examined was about 90%, and the results are represented in Figure 16.

The pass rate of its test results is only related to the DUT. By analyzing the test cases that did not pass, we may uncover two explanations for failed test cases by examining them:As illustrated in Figure 17, some of the overpass scenarios with mismatched space impair judgment and may cause the elevation to be misjudged;Some cars turn without signaling, which affects judgment.

### 4.3. CACC Test

CACC has evolved into an extension of ACC as Cooperative Intelligent Transport System (C-ITS) technology has advanced [24]. The cars in the queue may “see” the lead vehicle using V2X-CACC, allowing for a more thorough study of the fleet’s condition and decision-making. Because CACC is still developing, this framework includes a CACC test function for statistically evaluating the algorithm’s performance. with the circumstance of five cars, the following is an example of how this section of the test was performed. The leading car accelerates from 0 to 72 km/h and then maintains a constant speed, while the other vehicles follow the CACC algorithm as indicated in Figure 18.

CarTest captures vehicle driving data and terminates the test when a stable vehicle formation is formed. This paper presents a way for evaluating the CACC algorithm’s merits. The Table 3 lists some of the parameters.

To begin, the test logs are based on the simulation engine, which has a 0.01 s simulation step size. As a result, the logging module captures a collection of vehicle data at each step, which includes all of the fleet’s cars. Each test will last 100 s in total. The current timestamp is calculated using the simulation step duration and the current frame ID:(4)$$T_{m}=T\left(m\right)=0.01\times m.$$

The final average following distance may be calculated from the spacing between each vehicle at the termination point:(5)$$\bar{g_{M}}=\frac{1}{N-1}\sum_{i=1}^{N-1}g_{M,i}\;.$$

The average error is calculated based on the final average gap, as shown in Equation (Equation 6).
(6)$$\bar{E_{g}}=\frac{1}{N-1}\sum_{i=1}^{N-1}\left|g_{M,i}-\bar{g_{M}}\right|\;.$$

Based on the average error, it is determined whether the following distance reaches steady state. Therefore, error needs to be less than 0.001, as shown in Equation (Equation 7).
(7)$$F_{g}=\left\{\begin{array}{cc} 1, & \bar{E_{g}}<0.001\\ 0, & \bar{E_{g}}\geq 0.001. \end{array}\right.\;$$

In the same way, $F_{v}$ and $F_{a}$ can be calculated according to Equations (Equation 5)–(Equation 7). Based on $F_{g}$, $F_{v}$ and $F_{a}$, it is possible to determine whether the fleet has reached steady state, as shown in Equation (Equation 8).
(8)$$F=F_{g}\wedge F_{v}\wedge F_{a}\;.$$

The computation above is used to see if the steady state has been attained. It is deemed a failure if the steady state is not attained. If the steady state is obtained, the evaluation can proceed to the next phase. The necessary index parameters are listed in Table 4.

The average speed of each vehicle is calculated from 1-M for a total of M frames, as shown in Equation (Equation 9).
(9)$$\bar{v_{i}}=\frac{1}{M}\sum_{m=1}^{M}v_{m},i\;.$$

The standard deviation is calculated for a total of M frames of speed data for each vehicle, as shown in Equation (Equation 10).
(10)$$Sv_{i}=\sqrt{\frac{1}{M-1}\sum_{m=1}^{M}{\left(v_{m,i}-\bar{v_{i}}\right)}^{2}}\;.$$

The average of the standard deviation of the speed of all vehicles is taken as the overall standard deviation of that vehicle, as shown in Equation (Equation 11).
(11)$$Sv=\frac{1}{N}\sum_{i=1}^{N}\sqrt{\frac{1}{M-1}\sum_{m=1}^{M}{\left(v_{m,i}-\bar{v_{i}}\right)}^{2}}.\;$$

Similarly $S_{a}$ and $S_{g}$ can be calculated according to Equations (Equation 6)–(Equation 8).The time to reach steady state is not equal for each fleet, and the relevant parameters are shown in Table 5.

If the fleet as a whole reaches steady state, the time for all vehicles to reach the desired speed can be calculated, referring to Equation (Equation 4). After reaching steady state, the speed values are less than 0.001 error from the desired average speed, as shown in Equation (Equation 12).
(12)$$\forall x>m,|v_{m,i}-\bar{v_{M}}|<0.001\;$$
(13)$$TTS_{v}=T\left(\arg m\max f\left(m\right):=\left\{\left|v_{m,i}-\bar{v_{M}}\right|>0.001\right\}\right)\;$$

Similarly $TTS_{a}$ and $TTS_{g}$ can be calculated. The average of the three timestamps is calculated as the overall time stamp to reach steady state, as shown in Equation (Equation 14).
(14)$$TTS=\frac{TTS_{v}+TTS_{a}+TTS_{g}}{3}\;$$

As the measurement algorithm, the PID-based CACC algorithm [25] is employed. The tested CACC algorithm is suitably simplified and the parameters are shown in Table 6.

In CACC, V2X enables accurate knowledge of the movement of the vehicle in front of you, including the movement of the entire fleet. Here, P control $\left(G_{min}=5\;m,\;T_{g}=1\;s\right)$ is used to control the host vehicle acceleration, as shown in Equation (Equation 15).
(15)$$a_{g}^{*}=K_{a}\left(a_{f}-a\right)+K_{v}\left(v_{f}-v\right)+K_{g}\left(g-G_{min}-vT_{g}\right).\;$$

The host vehicle adjusts its acceleration according to the difference in speed, acceleration and interval between the host vehicle and the vehicle in front of it. When the speed, acceleration, and distance between the host vehicle and the preceding vehicle are constant, then the acceleration adopted by the host vehicle is 0. When the acceleration of all vehicles in the convoy is 0, then the entire convoy reaches steady state and has the same speed as well as the gap, where $K_{v},K_{a},K_{g}$ is the control factor of each, which takes different values depending on the unit and importance. In the P model, the value of each control factor will directly affect the performance of the model.

For four experiments, various parameters ($K_{v},K_{a},K_{g}$) are used listed in Table 7. We will keep track of each vehicle’s whole state change. The results of these four experiments can be tested and evaluated by CarTest. The measured algorithm is not limited to PIs, we evaluate only the fleet status.

P control of acceleration is not used in the P1 and P2 models, therefore their $k_{a}=0$. The CACC control algorithm for P1 and P2 is shown in Equation (Equation 16).
(16)$$a_{g}^{*}=K_{v}\left(v_{f}-v\right)+K_{g}\left(g-G_{min}-vT_{g}\right).\;$$

The state curves of experiment P1 and experiment P2 were more fluctuant and took longer to reach the steady state. It is obvious that the performance of these two groups of algorithms is poor, as shown in Figure 19.

The CACC models for P3 and P4 add control of acceleration, are shown in Equation (Equation 15). The experimental results of P3 and P4 were significantly better than those of P1 and P2, as shown in Figure 20.

The state curves of P3 and P4 are closer, and it is difficult to distinguish the performance with the naked eye. Combining the foregoing assessment methodologies yielded a more detailed examination of the experimental data. The Table 8 lists the exact experimental settings as well as the assessment outcomes.

The process of calculating the assessment results is referred to Equations (Equation 4)–(Equation 14). The total score formula:
(17)$$Score=300-2\times S_{v}-S_{a}-10\times S_{d}-\bar{g_{m}}-\bar{v_{m}}-TTS.$$

P3 and P4 are in charge of acceleration, and their time to steady state is substantially shorter, resulting in superior performance. In conclusion, while comparing various control settings, the P4 algorithm is more suited. Its P4 score has increased by 9.15% over its P3 level. The tests above demonstrate that the test framework can do CACC-related testing.

### 4.4. Large-Scale Test

Vehicle popularity is steadily expanding as technology advances, and more and more cars are passing on the road. As more cars are equipped with OBU, their capacity to communicate with a large number of terminals must be further investigated [26]. As a result, large-scale testing is becoming increasingly critical, but there is no suitable HIL platform to support large-scale testing.This work proposes a suitable testing system.

The platform may handle 10–160 OBU as background OBU using CarTest for large-scale testing, allowing the components under test to execute communication tests in a large number of OBU settings and assess their packet loss rate, warning accuracy, and so on. Meanwhile, large-scale testing provides a laboratory testing option, allowing for the installation of 80 OBUs indoors. At the same time, background cars with eight OBUs for outdoor testing are being developed, as shown in Figure 21 and Figure 22.

The involved OBU which were being tested supports GPS/QZSSL1C/A&L5, BDSB1I and GALE1&E5a. The OBU is equipped with IMU as well as support for RTK technology, with positioning accuracy up to centimeter level. For indoor testing, GNSS antenna amplifiers need to be placed in the lab, as shown in Figure 11. The test will start after ensuring that all OBUs as well as DUTs receive the information.

All OBUs can quickly and accurately acquire a position through the indoor GPS amplifier.

Background OBU (BOBU, ID:1-160) and DUT logs are kept during the test. The logs contain the received BSM as well as communication quality data (CBR: channel busy rate, PER: packet error ratio, RSRP: reference signal received power, etc.). We will further analyze the important settings based on the log data. As indicated in Figure 23, we tested 10–160 OBU.

There is no denying that, as the number of OBUs grows, so does the packet error ratio. The PER profile varies greatly when the number of background OBUs is 160. The maximum percentage of packet loss is 26.48%, while the lowest rate is 1.62%. The analysis is displayed in Figure 24 when combined with the channel busy rate.

To get the following graph, the CBR of all BOBU is averaged and merged with the PER of DUT. The PER of the DUT is positively connected with the CBR of the BOBU, as shown in Figure 25.

CarTest has completed the testing and evaluation of the large-scale OBU test, bridging the gap of large-scale testing of existing HIL test platforms.

### 4.5. GNSS Test

Due to cold start, the OBU may not be able to be located effectively when scene switching is conducted.

Cold start is the process of starting up in an unfamiliar environment until contact is made with the surrounding satellites and coordinates are calculated;Hot start refers to when there is not much movement in the location where it was last shut down, but the time from the last positioning must be less than 2 h. The last estimated visible satellite’s position is saved;

We can test the cold and hot start performance of GNSS chips for OBU using CarTest. We have put together a test case library for several locales, some of which are included in Table 9.

Figure 26 is a flow chart for the exam. If the site is still operational after 15 min, it is deemed a failure.

The hardware device is shown in Figure 27.

As illustrated in Figure 28, the cold start may have failed to start, and the average speed is much slower than the hot start.

When automating tests, changing test cases produces a location jump, resulting in a “cold start”, as seen in Figure 29.

This position change results in a considerable increase in positioning time, and it may even need a reboot to go back to normal. There are numerous options for dealing with this problem:Make all test cases’ starting points the same (X,Y). Return to the beginning of a scenario once it has completed. The following scenario’s starting point is also (X,Y), eliminating the location leap, as shown in Figure 29;To maintain continuity, set up distinct test cases in various areas of a test scenario, as shown in Figure 30.

Both alternatives have drawbacks. When contrasting genuine situations, the first scheme is unable to unify the starting point. On the scene’s road, the second plan is more challenging. As a result, the first strategy may be used to imaginary roads in general. The second method, for example, Mcity, can be utilized for genuine test sites.

## 5. Conclusions

V2X apps are still in their early stages of development and may mislead drivers, putting road safety at risk. However, field testing presents several challenges. As a result, we present in this study a hardware-in-the-loop simulation-based testing framework that simplifies application development, testing, and algorithm performance comparison. The testing framework includes a large library of test cases to cover a wide range of testing needs, as well as test logging and data visualization. We compiled a list of V2X-related applications, broadly dividing them into two categories: early warning and collaborative.

Taking ICW testing as an example, the performance of the algorithm can be analyzed by CarTest to obtain the overall pass rate and to locate test failure cases to help engineers improve the security of the algorithm. We also conducted CACC tests to provide a method to evaluate its performance. The method can be used to evaluate the advantages and disadvantages of multiple algorithms. After large-scale testing, the communication of OBU in complex channel environments can be counted and analyzed. When the number of OBU is greater than 160, the PBR is about 12% and it is positively correlated with CBR. Finally, the positioning test of OBU is an easier part of HIL testing to ignore. After analyzing the time consumed by OBU’s hot and cold starts, this paper proposes two better solutions to solve the location jump problem in HIL testing.

It demonstrates that the platform can do the necessary tests. The platform ran 1160 test scenarios over the course of 14 h. Only 40 test scenarios may be tested in 4 h during the real road test. As a result, CarTest can increase testing efficiency by 8.2 times. This study outlines the major issues with V2X-HIL and suggests remedies for researchers to consider. The flaws of several application algorithms, such as elevation judgment, are also highlighted after testing.

The creation of test cases still requires a significant amount of manual labor. To automate the test case generation, we are exploring employing reinforcement learning or neural networks. We intend to improve the test case library and modify the assessment process in the future. We can achieve a more accurate outcome evaluation by capturing additional data. 

## Figures and Tables

**Figure 1 sensors-22-05019-f001:**
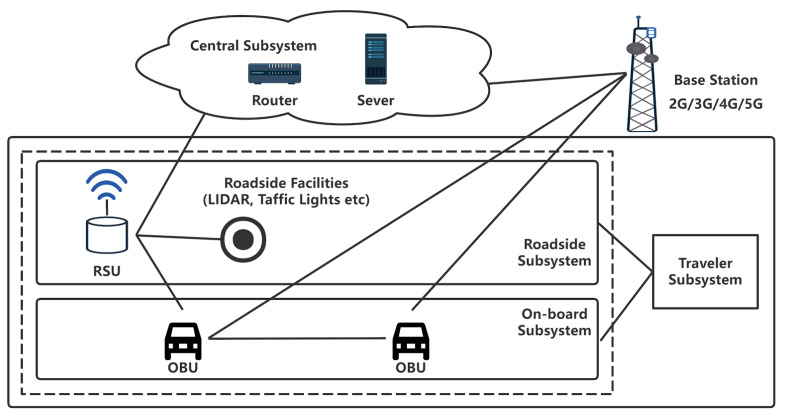
ITS system model; vehicles can obtain more accurate information.

**Figure 2 sensors-22-05019-f002:**
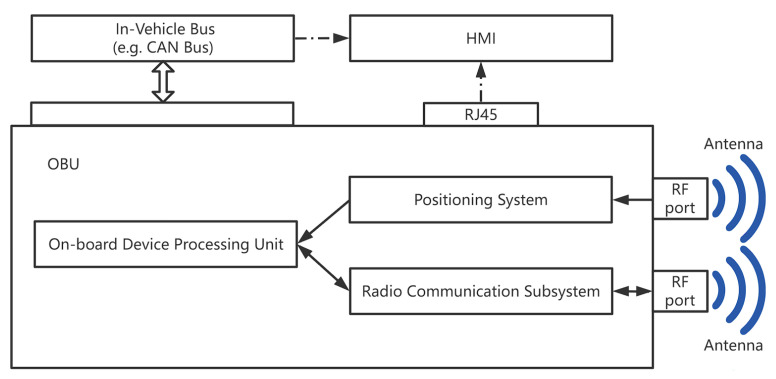
Basic components of the OBU.

**Figure 3 sensors-22-05019-f003:**
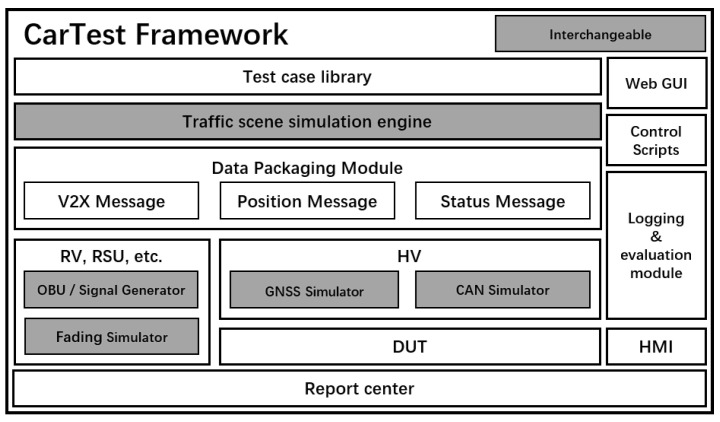
CarTest system model.

**Figure 4 sensors-22-05019-f004:**
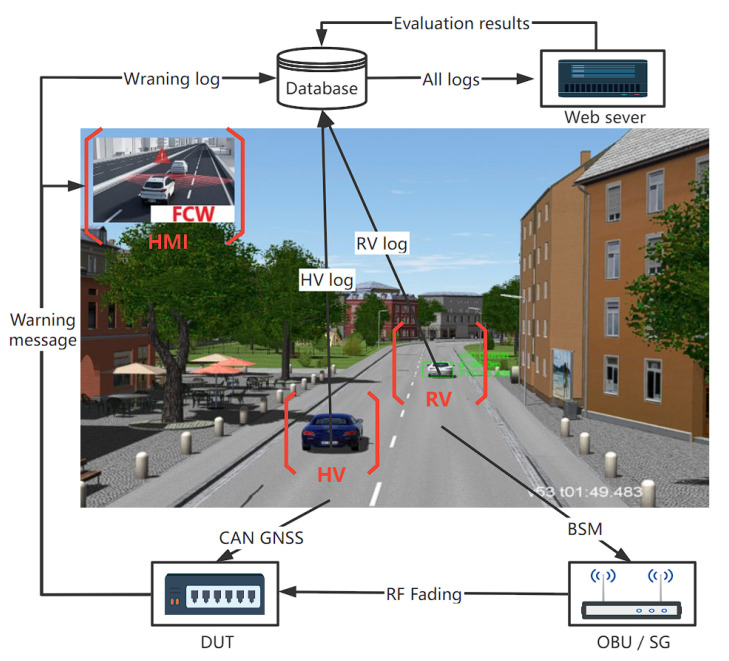
CarTest data flow chart.

**Figure 5 sensors-22-05019-f005:**
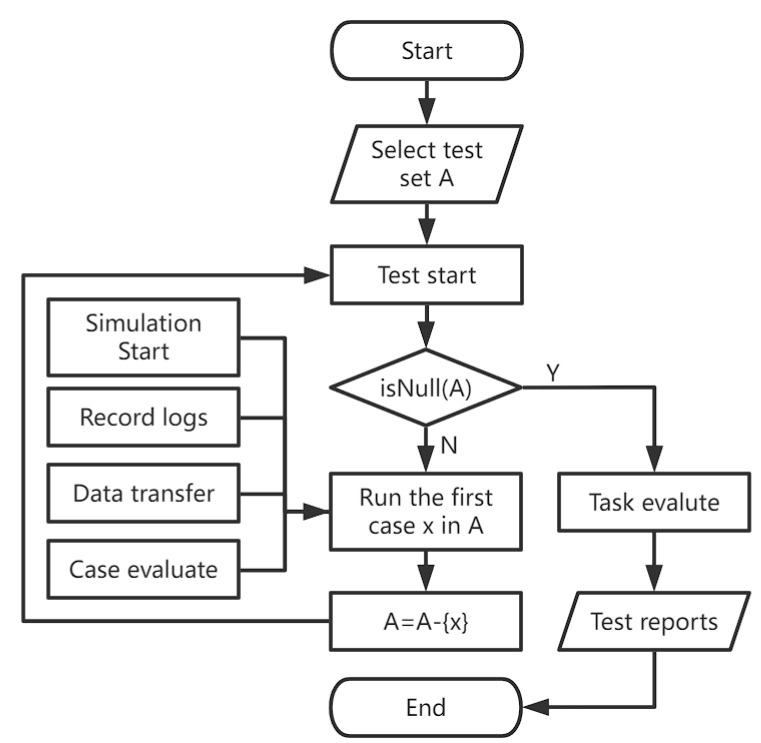
CarTest test flow chart.

**Figure 6 sensors-22-05019-f006:**
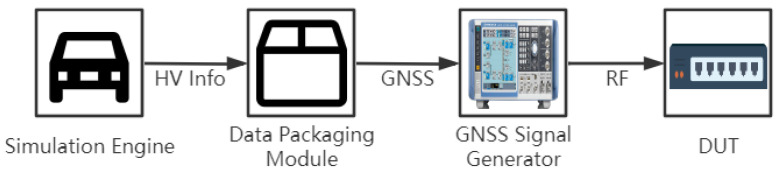
GNSS simulation hardware connection schematic.

**Figure 7 sensors-22-05019-f007:**
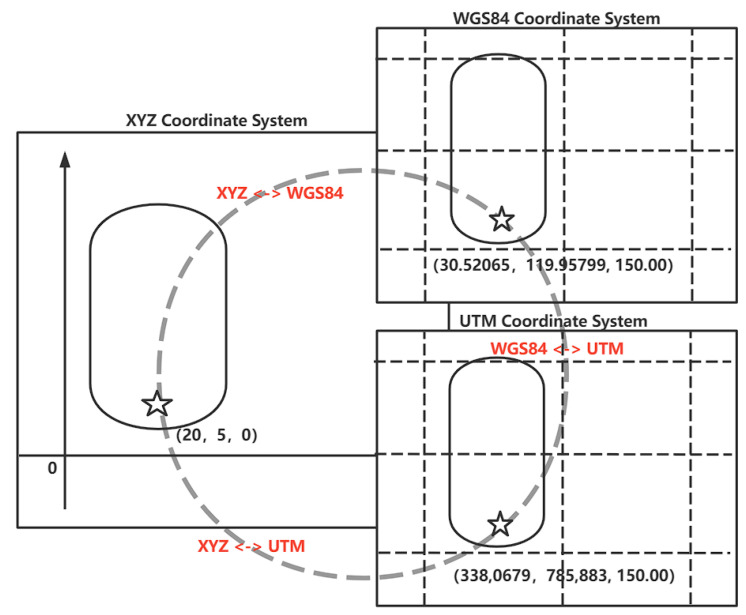
Transformation of coordinates, the star represents a point in the map.

**Figure 8 sensors-22-05019-f008:**
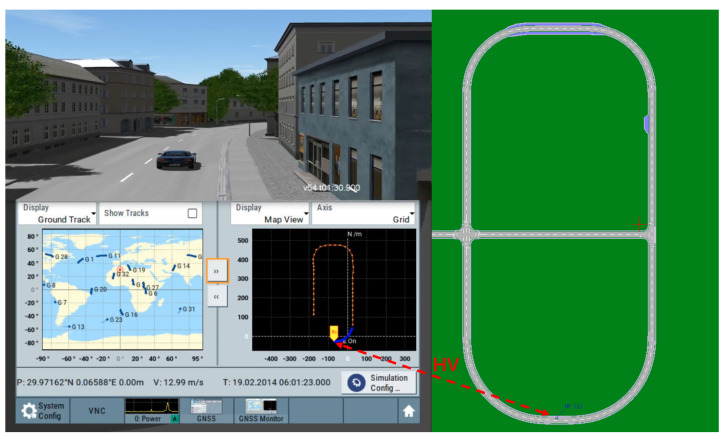
GNSS simulation in VTD simulator.

**Figure 9 sensors-22-05019-f009:**
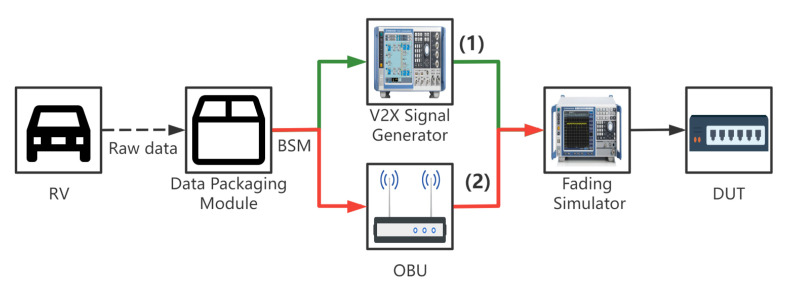
Hardware architecture diagram of V2X simulation test: (1) V2X signal generator, (2) OBU as signal generator.

**Figure 10 sensors-22-05019-f010:**
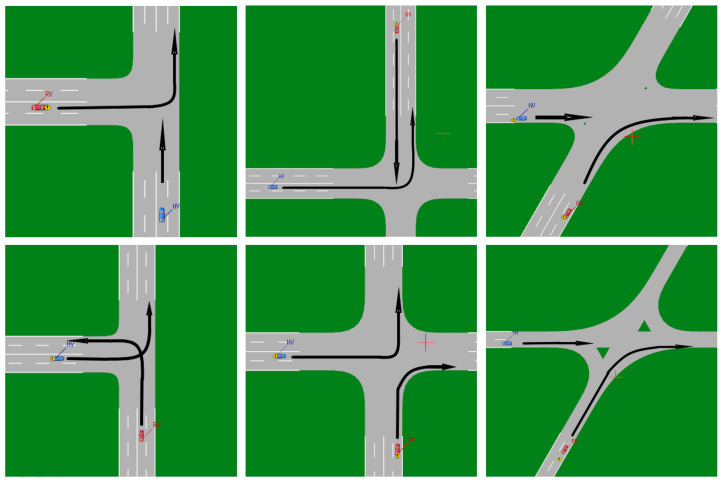
ICW test cases in VTD simulator.

**Figure 11 sensors-22-05019-f011:**
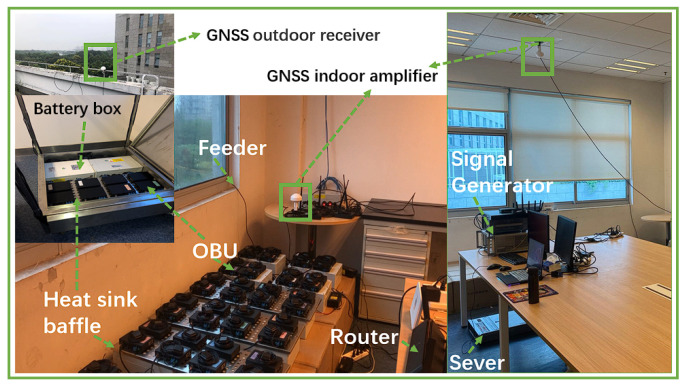
Indoor testing setup.

**Figure 12 sensors-22-05019-f012:**
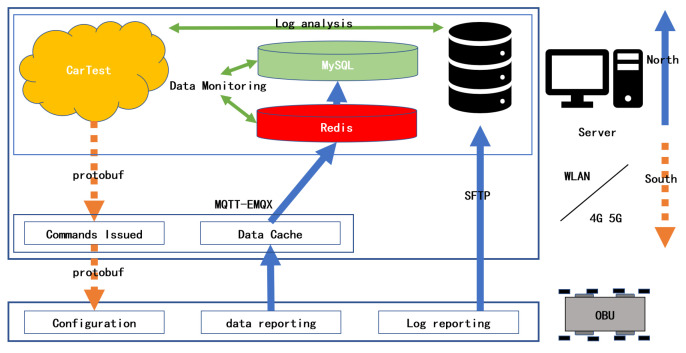
Indoor test data flow and software components.

**Figure 13 sensors-22-05019-f013:**
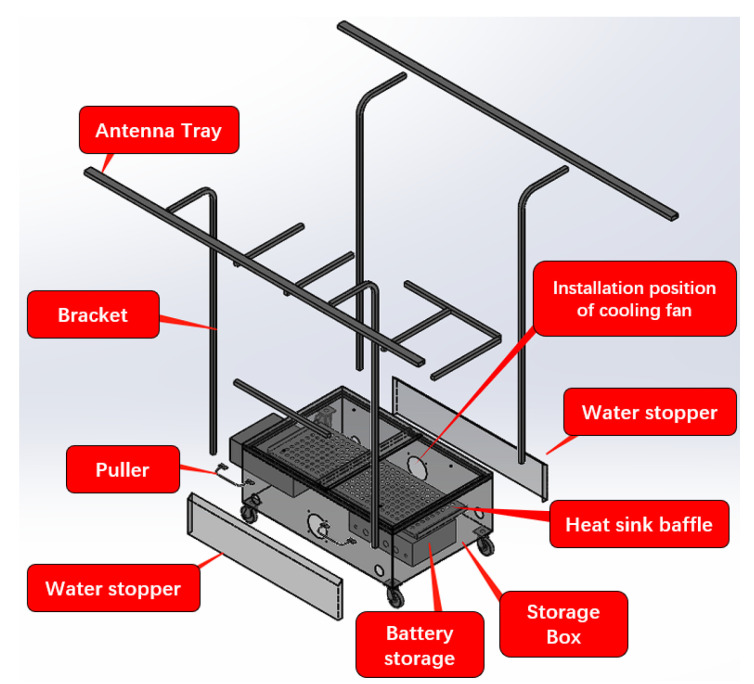
Design of outdoor test trolley.

**Figure 14 sensors-22-05019-f014:**
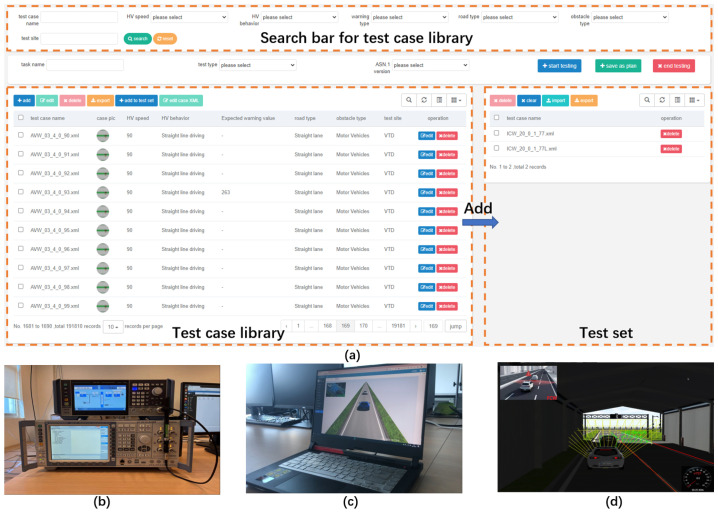
(**a**) Test case administration interface (**b**) Signal generators and server (**c**) Client display in safe driving situations (**d**) Client-side display of forward collision warning and sensor effect.

**Figure 15 sensors-22-05019-f015:**
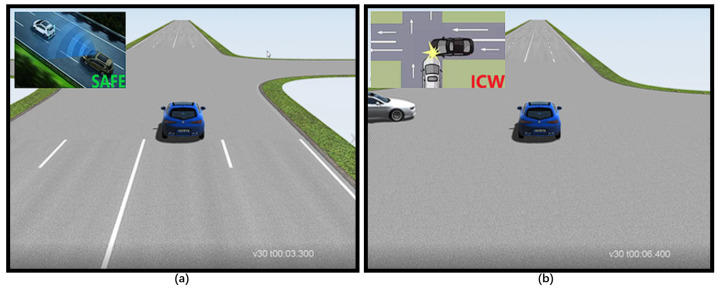
FCW test: (**a**) safe at 3.3 s, (**b**) warning at 6.4 s.

**Figure 16 sensors-22-05019-f016:**
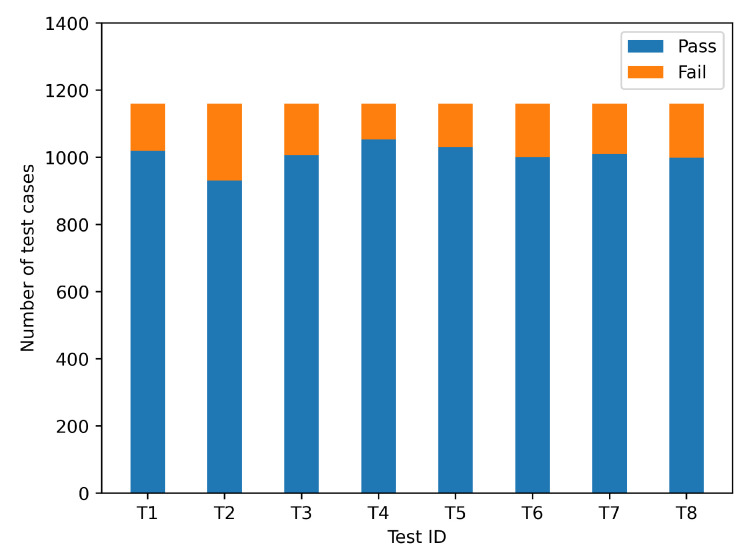
Comparison of ICW test pass rates with 1160 cases.

**Figure 17 sensors-22-05019-f017:**
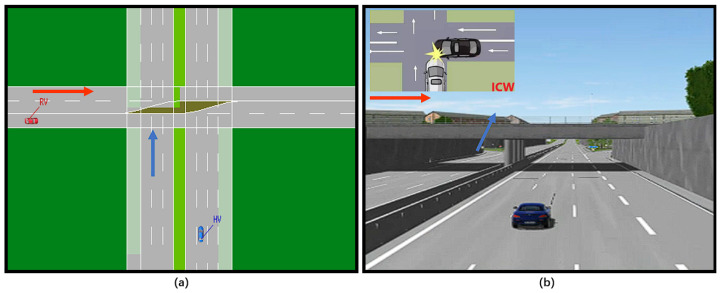
ICW test: (**a**) Top view, (**b**) Main view with unsuitable warning in VTD simulator.

**Figure 18 sensors-22-05019-f018:**
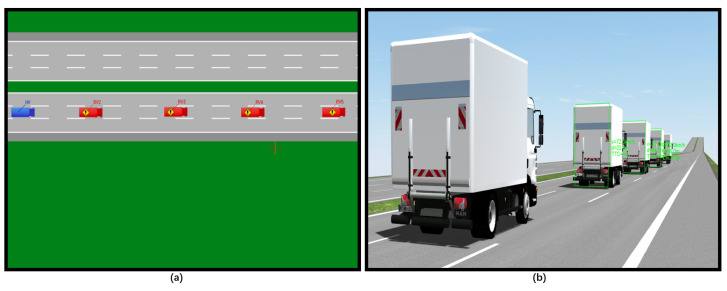
CACC test: (**a**) Top view of a CACC case, (**b**) Main view of a CACC case in VTD simulator.

**Figure 19 sensors-22-05019-f019:**
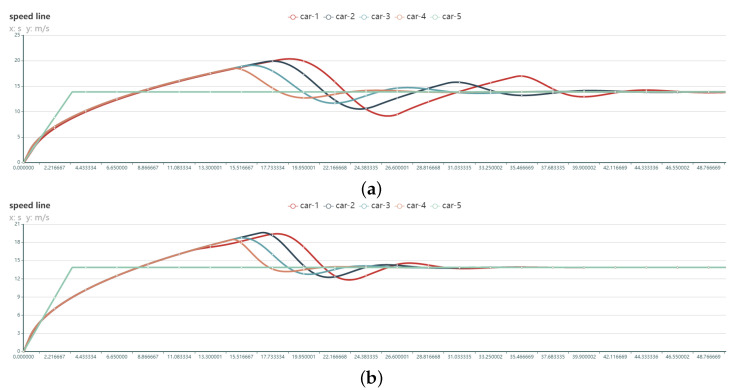
Speed variation curves of 5 vehicles in the convoy using different PID model control (no acceleration control): (**a**) P1 $\left[K_{v}=0.2,K_{g}=0.1\right]$ (**b**) P2 $\left[K_{v}=0.2,K_{g}=1.0\right]$.

**Figure 20 sensors-22-05019-f020:**
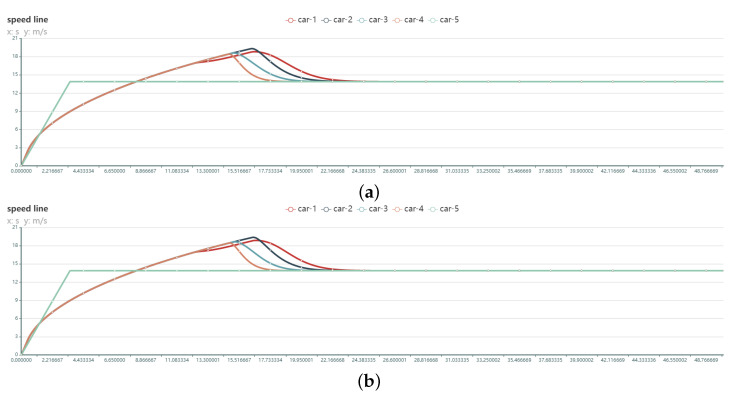
Speed variation curves of 5 vehicles in the convoy using different PID model control (with acceleration control): (**a**) P1 $\left[K_{v}=1.0,k_{a}=0.8,K_{g}=4.0\right]$ (**b**) P2 $\left[K_{v}=0.75,k_{a}=0.7,K_{g}=4.125\right]$.

**Figure 21 sensors-22-05019-f021:**
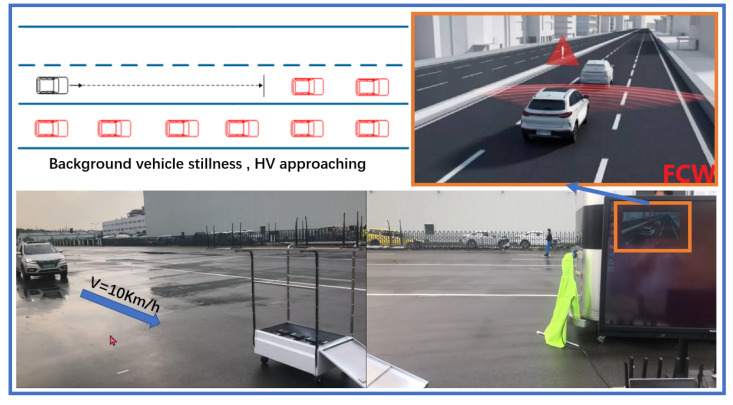
Outdoor Large-scale Test with HV and BOBUs.

**Figure 22 sensors-22-05019-f022:**
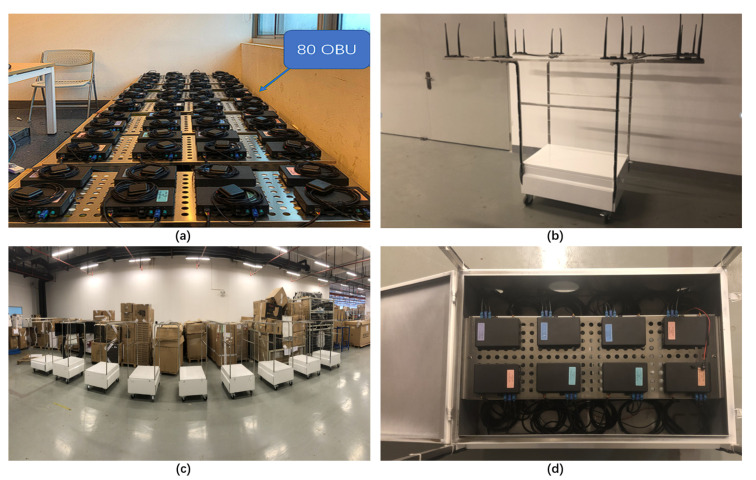
(**a**) 80 OBU for indoor testing, (**b**) background vehicle, (**c**) 8 background vehicles, (**d**) Box of the vehicle and 8 OBUs.

**Figure 23 sensors-22-05019-f023:**
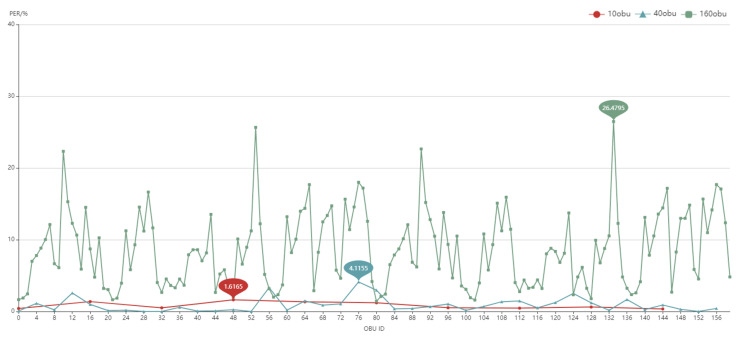
Large-scale test result (PER with 10,40,160 BOBU).

**Figure 24 sensors-22-05019-f024:**
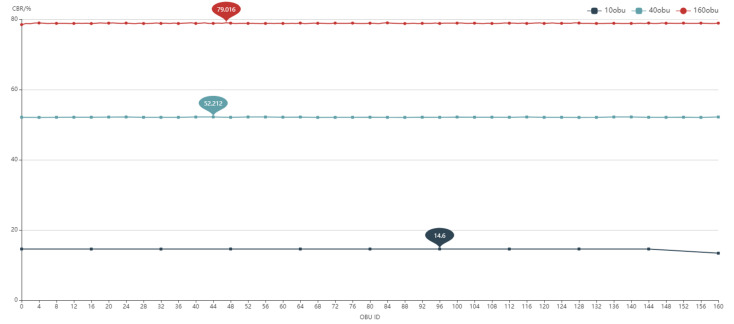
Large-scale test result (CBR with 10,40,160 BOBU).

**Figure 25 sensors-22-05019-f025:**
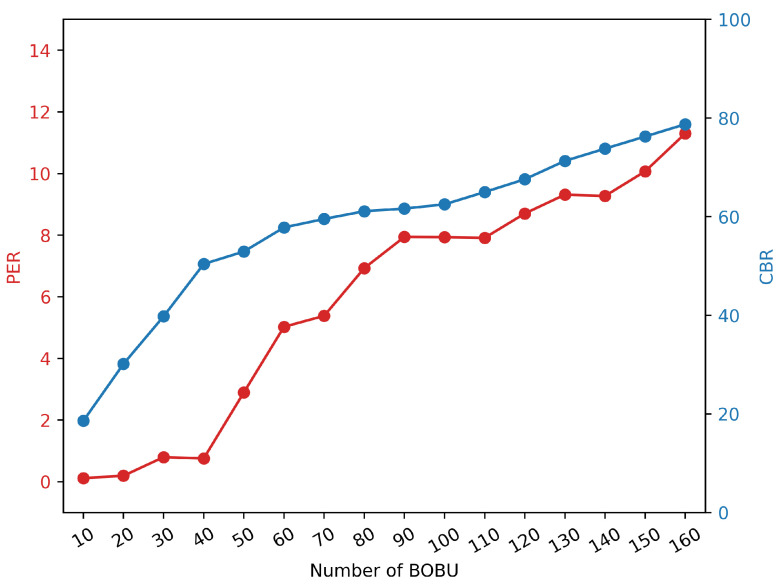
Comparison of CBR and PER with 10-160 BOBU.

**Figure 26 sensors-22-05019-f026:**
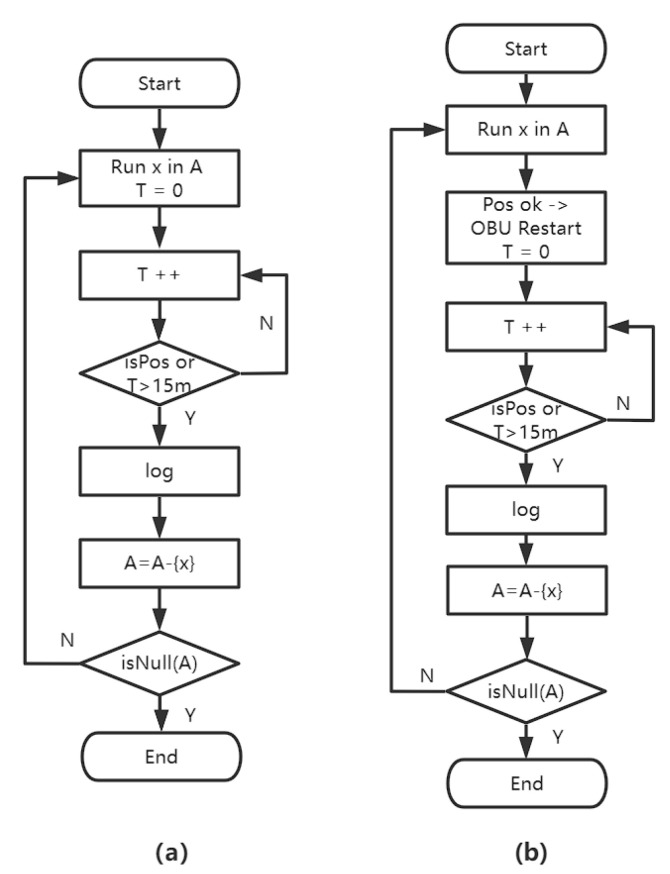
GNSS test: (**a**) Cold start test, (**b**) Hot start test.

**Figure 27 sensors-22-05019-f027:**
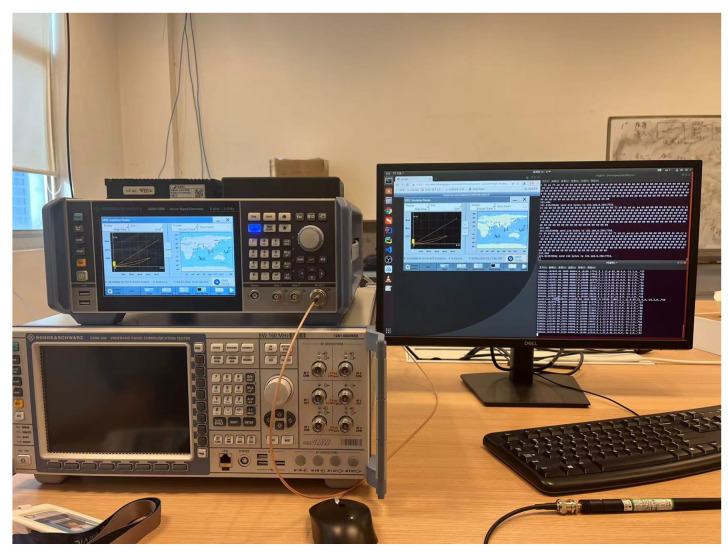
GNSS test hardware and test situation.

**Figure 28 sensors-22-05019-f028:**
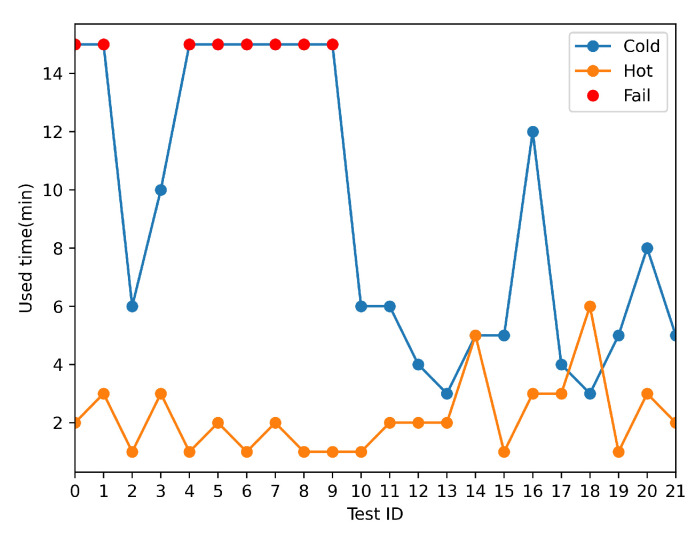
GNSS cold and hot start test result.

**Figure 29 sensors-22-05019-f029:**
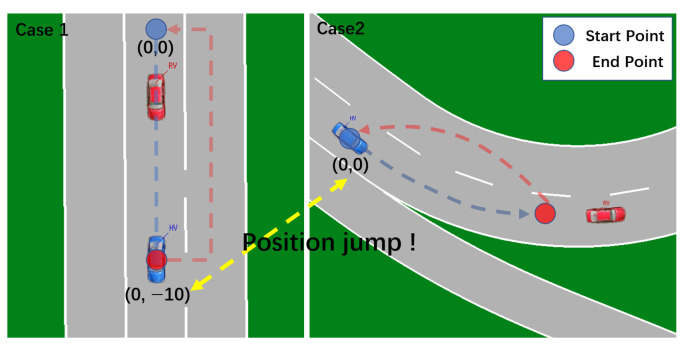
Test case switching causes OBU “Cold start” and solution in VTD simulator.

**Figure 30 sensors-22-05019-f030:**
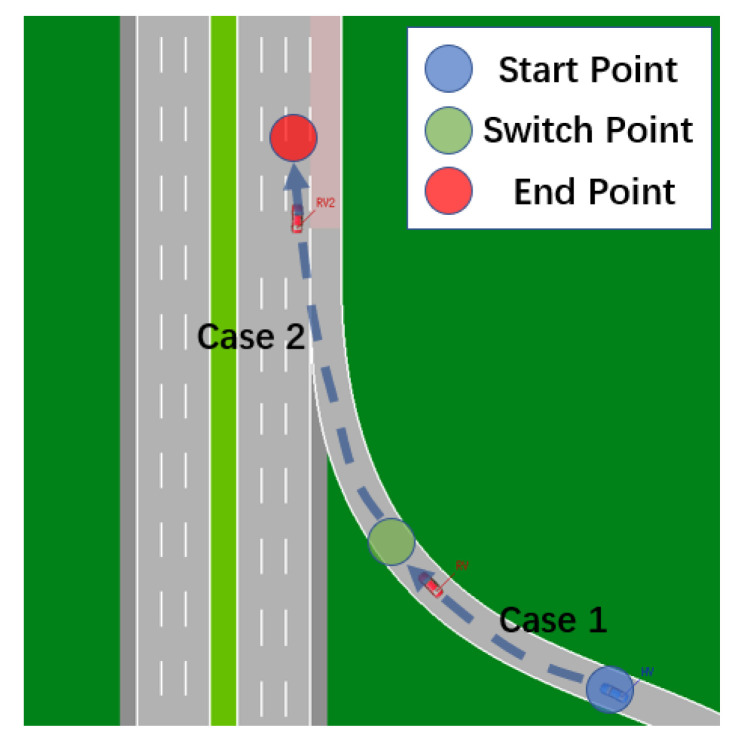
Road scenarios for continuity testing in VTD simulator.

**Table 1 sensors-22-05019-t001:** CAN .dbc example.

Message	Unit	Minimum	Maximum
TransmissionState	-	0	15
ABS Active	-	0	1
Traction Control Active	-	0	1
Brakes Active	-	0	1
Panic brake active Hard Braking	-	0	1
Longitudinal Acceleration	m/s${}^{2}$	−15.36	15.33
Steering Wheel Angle	degree	−2048	2047.88
Vehicle Speed	km/h	0	511.984
LF Wheel Speed	km/h	0	511.969
RF Wheel Speed	km/h	0	511.969
LR Wheel Speed	km/h	0	511.969
RR Wheel Speed	km/h	0	511.969
Left Turn Signal	-	0	3
Right Turn Signal	-	0	3
hazard lights on	-	0	1
fog lights on	-	0	1
LF Wheel RPM	-	−32,768	32,767
RF Wheel RPM	-	−32,768	32,767
LR Wheel RPM	-	−32,768	32,767
RR Wheel RPM	-	−32,768	32,767

**Table 2 sensors-22-05019-t002:** V2X applications.

Category	Full Name	Simple Name
V2V	Forward Collision Warning	FCW
V2V/V2I	Intersection Collision Warning	ICW
V2V/V2I	Left Turn Assist	LTA
V2V	Blind Spot Warning-Lane Change Warning	BSW-LCW
V2V	Do Not Pass Warning	DNPW
V2V-Event	Emergency Brake Warning	EBW
V2V-Event	Abnormal Vehicle Warning	AVW
V2V-Event	Control Loss Warning	CLW
V2I	Hazardous Location Warning	HLW
V2I	Speed Limit Warning	SLW
V2I	Red Light Violation Warning	RLVW
V2P/V2I	Vulnerable Road User Collision Warning	VRUCW
V2I	Green Light Optimal Speed Advisory	GLOSA
V2I	In-Vehicle Signage	IVS
V2I	Traffic Jam Warning	TJW
V2V	Emergency Vehicle Warning	EVW
V2I	Vehicle Near-Field Payment	VNFP

**Table 3 sensors-22-05019-t003:** Notations of CACC evaluation (1).

Variables	Notations
$T_{m}$	Timestamp of the mth frame
$F_{g}$	Flag of steady gap
$F_{v}$	Flag of steady speed
$F_{a}$	Flag of steady acceleration
*F*	Flag of steady status
$g_{m,i}$	In frame m, the gap of the ith car
$v_{m,i}$	In frame m, the speed of the ith car
$a_{m,i}$	In frame m, the acceleration of the ith car
$\bar{g_{m}}$	In frame m, average gap
$\bar{v_{m}}$	In frame m, average speed
$\bar{a_{m}}$	In frame m, average acceleration
*N*	Total number of vehicles
*M*	End frame ID

**Table 4 sensors-22-05019-t004:** Notations of CACC evaluation (2).

Variables	Notations
$\bar{v_{i}}$	Average speed of the ith vehicle
$\bar{a_{i}}$	Average acceleration of the ith vehicle
$g_{i}$	Average gap of the ith vehicle
$Sv_{i}$	Standard deviation of the speed of the ith vehicle
$Sa_{i}$	Standard deviation of the acceleration of the ith vehicle
$Sg_{i}$	Standard deviation of the gap of the ith vehicle
Sv	Overall speed standard deviation
Sa	Overall acceleration standard deviation
Sg	Overall gap standard deviation

**Table 5 sensors-22-05019-t005:** Notations of CACC evaluation (3).

Variables	Notations
$TTS_{v}$	Times for speed to reach steady state
$TTS_{a}$	Times for acceleration to reach steady state
$TTS_{g}$	Time for gap to reach steady state
TTS	Overall time for reaching steady state

**Table 6 sensors-22-05019-t006:** Notations of PID-based CACC algorithm.

Variables	Notations
$a_{f}$	Acceleration of the car in front
*a*	Acceleration of HV
$a_{g}^{*}$	Acceleration strategies of HV
$v_{f}$	Speed of the car in front
*v*	Speed of HV
*g*	Gap with the front car
$G_{min}$	Minimum safety gap
$T_{g}$	Expected time gap
$K_{a}$	Scale factor of acceleration
$K_{v}$	Scale factor of speed
$K_{g}$	Scale factor of gap

**Table 7 sensors-22-05019-t007:** Parameter setting for CACC test.

	P1	P2	P3	P4
$K_{v}$	0.200	0.200	1.000	0.750
$K_{a}$	-	-	0.800	0.700
$K_{g}$	0.100	1.000	4.000	4.125

**Table 8 sensors-22-05019-t008:** CACC test results and evaluation indicators.

	P1	P2	P3	P4
$S_{v}$	2.127	1.951	1.891	1.894
$S_{a}$	0.747	0.710	0.671	0.674
$S_{g}$	4.109	3.465	3.410	3.411
$\bar{g_{m}}$	14.889	15.889	15.889	15.889
$\bar{v_{m}}$	13.889	13.889	13.889	13.889
TTS	74.911	46.011	43.494	26.250
$TTS_{v}$	70.150	44.500	26.833	25.967
$TTS_{a}$	81.433	49.783	77.167	27.233
$TTS_{g}$	73.150	43.750	26.483	25.550
Score	150.221	184.952	188.180	205.402

**Table 9 sensors-22-05019-t009:** CACC test result.

Moving Track	Longitude	Latitude
Barcelona	2.23817 N	41.40908 E
Melbourne	37.80819 S	144.96783 E
Tokyo	35.66667 N	139.77492 E
Munich	48.14550 N	11.57856 E
NewYork	40.75957 N	73.98498 W
Nuerburgring	50.33275 N	6.93630 E
Shanghai	31.230416 N	121.473701 E
Beijing	39.904211 N	116.407395 E
Nepal	28.394857 N	84.124008 E
Stockholm	59.329323 N	18.068581 E

## Data Availability

Not applicable.

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
