# Peer review of "A Hardware-in-the-Loop V2X Simulation Framework: CarTest"

_sensors, 2022, doi:10.3390/s22135019_

Round 1

Reviewer 1 Report

Dear Authors,

I have received perceived your paper as well organized and well written. Your approach may be of use for other researchers in the field using V2X technologies.

However, there are some remarks and missing data that I would like to point out. If answered, it will bring clarity and increase quality of the paper.

- the software you tested and mentioned in the paper is it your own software development  or a commercial one? I was not able to find an answer myself for this matter. Please add a paragraph to clarify the software origin in the manuscript.

- I have seen the images in the manuscript - Fig 8., 10, 14, 15, 25, 26 with the green background. I have see scenarios done with Carla or Sumo simulators. Please introduce some details in images caption regarding what software was used to elaborate these  road scenarios.

- Please specify in the text the hardware precision of GNSS  regarding positioning of the involved OBU which were being tested (Fig. 17, 18).

- You have mentioned laboratory tests (fig 18, 19). Were all OBUs able to receive a valid position from the satellite inside the facility during tests?  Please add a paragraph with this answer. It will validate your indoor tests.

- What interface have you used to collect and analyse data (fig. 18, 19) and logs ? Have you used wireless LAN, batteries pack to supply power? I have seen that are no cables involved, at least from the angle of where the pictures have been taken. Or the data was collected manually? Please add a paragraph to clarify.

- Please add more description for the UTM coordinate system.

- please add details, mathematical description to the transformation of coordinates involved xyz -> Utm

- Figure 16 is not clear even with magnification. please replace it with a higher resolution and thicker lines. You may also split it in two for P1 and P2, and the other one with P3 and P4.

- A major downside of the article is that there is no description of the setup involved, hardware, OBU, GNSS testing setup, location of the facility where the tests were conducted. Keep in mind the fact you want to bring in the field reproducible and scalable tests. It would bring relevance to the paper if you were to introduce a chapter for setup regarding the conducted field testing and indoor testing.

- The conclusion must be extended to underline the tests involved: I remember the following - ICW test Software test with hardware in the loop; - CACC test, Large-scale test (PER, DUT), GNSS tests cold, hot, jumps and solutions. Please extend the conclusion chapter and organize the ideas to bring clarity.

I am looking forward to the updated version of the manuscript. I am sure that the additional effort will bring more value and visibility to your research.

Reviewer 2 Report

This paper presents a study for a hardware-in-the-loop simulation-based testing framework that simplifies application development, testing, and algorithm performance comparison. The testing framework includes a large library of test cases to cover a wide range of testing needs, as well as test logging and data visualization. However, the quality of presentation of the methods of the paper is quite deficient. Some specific comments are as follows:

-Μany abbreviations are not spelled correctly (e.g. HIL). It is not obvious.

-Α better analysis is required for the simulation of Inter-Vehicle Communication (IVC). What kind of channel fading simulator is used? In addition, the RF signals face phenomena such as strong interference or RF jamming. How are these phenomena dealt with in the proposed model?

-Is the proposed HIL a real-time system?  Ηow many cars they are able to simulate in real time? How are  real time droppes treated in wireless communication between simulated vehicles and DUT?

-Does the proposed HIL contains the upper layers (e.g. for the HV and RV)? For example, which is the MAC layer for the transmission of packets between the simulated vehicles and the DUT?Ιs  it adaptable to heterogeneous communication technologies?

-All the Figures of this paper is of poor quality. Especially,  the Figure 16. What is (P1,P2,P3,P4)? I can imagine that are the reward functions of the PID-based CACC algorithm (reference 25), but they are not mentioned anywhere.

Round 2

Reviewer 1 Report

Dear Authors,

I am confident that the paper has improved from the previous version. Also, there are still some aspects that you have not addressed yet. I present those issues below:

1. CACC detailed abbreviation is only present in the abstract. It is not present in the text.

2. ICW detailed abbreviation is only present in the abstract. It is not present in the text. Is present only in table 2

3. You have not addressed properly the issue below . While the added paragraph lines 123-132 is give some remarks. It does not bring enough clarity to the reader. The introduced example between 141-146 lines is not clear. I see three types of coordinates XYZ, WGS84 and UTM. Some transforming equations would help much and also description of the system of coordinates and origin references. Please take some time to add a proper description with equations. Keep in mind that this is that part that relates to the investigated cold and hot start of the OBUs.

    3. Point 7: please add details, mathematical description to the transformation of coordinates involved
xyz -> Utm.
Response 7: We will add details, mathematical description to the transformation of coordinates
involved xyz -> Utm.

4. The previous presented issue below has not been answered, but not addressed in the manuscript.

Point 4: You have mentioned laboratory tests (fig 18, 19). Were all OBUs able to receive a valid
position from the satellite inside the facility during tests? Please add a paragraph with this answer.
It will validate your indoor tests.
Response 4: Yes, all OBUs can receive a valid position from the satellite inside the facility during
tests. We placed an outdoor antenna on the roof of the building and connected it to the interior
through a GPS amplifier. We will also add a paragraph to describe our approach in detail. No added paragraph was found.

5. Figure 16 (P1, P2, P3, P4) have not been slip as suggested and answered. The quality of the image has not changed. Please replace with two images with higher quality and with detailed caption to capture the presented aspects.

Point 8: Figure 16 is not clear even with magnification. please replace it with a higher resolution and
thicker lines. You may also split it in two for P1 and P2, and the other one with P3 and P4.
Response 8: I am very sorry for the oversight regarding the quality of this image. We will export a
clearer picture and split it in two for P1 and P2, and the other one with P3 and P4.

6. The experiments noted P1, P2, P3, P4 for CACC have a poor description. There are missing aspects revolving what is wanted to be found and achieved in these experiments.

7. HIL detailed abbreviation is not present in the manuscript or abstract. Add a detailed abbreviation to add readability.

8. Quality of the images 1, 2, 5, 9, 12, 14, 19, 22, 24, 25. Take the time to remake them and add a proper captioning. Figure captions are mission or are to short. Add detailed axes description, abbreviations, scope if needed. For example, Figure 19 has no description.

Please take into account the above presented issues and recommendations. I tried to be brief and also direct. I am sure that the added work will pay off. I am looking forward to your final submission.

Best regards

Author Response

Dear Editors and Reviewers:

Thank you very much for your constructive comments and suggestions, which have helped us a lot to improve the quality of the paper.

Reviewer 2 Report

The paper has improved considerably. Most of the reviewers' comments were resolved and several points that were difficult to clarify were clarified.

Author Response

Dear Editors and Reviewers:

Thank you very much for your constructive comments and suggestions which would help us both in English and in depth to improve the quality of the paper. For the issues that still need improvement, we tried our best to improve the manuscript and made some changes in the manuscript. We appreciate for Editors/Reviewers’ warm work earnestly, and hope that the correction will meet with approval.

Round 3

Reviewer 1 Report

Dear Authors,

I agree that previous points 1, 2 and 3 have been answered.

1. Just to be clear I present your answer below for point 4.

You have mentioned laboratory tests (fig 18, 19). Were all OBUs able to receive a valid
position from the satellite inside the facility during tests? Please add a paragraph with this answer.
It will validate your indoor tests.
Response 4: We describe it in the lab test setup section (Chapter 3.6), but it may not be described
clearly. We will emphasize this again at picture 18.

No emphasis is present at picture 18 in the present manuscript.

2. A appreciate that you took into account to split figure 19. However, the lines, axes and text are too small and thin. You may enlarge on the vertical axis. Remake the figures 19 and 20 with thicker lines.

3. Also, you need to introduce the score equation used to obtain presented values in table 8.

4. I still can not grasp the conducted experiments P1, P2, P3, P4. I see the parameters in the table 7 Kv, Ka and Kd. An additional paragraph on the scope and meaning of each of these experiments must be added to bring clarity. I am unable to distinguish the experiments P1, P2, P3, P4 or evaluate them. Again figures 19,20 are not clear, have a poor caption description and thus does not support the text presented between line 344-351.

5. Point 7 has been answered.

6. Figure 2 has no reference in the text

Overall, the manuscript has improved. Reply to the comments above to get ready to be published.

Best regards

Author Response

(The authors gave the same response as above.)
